# NPAS1-ARNT and NPAS3-ARNT crystal structures implicate the bHLH-PAS family as multi-ligand binding transcription factors

Dalei Wu[1][†], Xiaoyu Su[1], Nalini Potluri[1], Youngchang Kim[2], Fraydoon Rastinejad[1]*

[1]Integrative Metabolism Program, Sanford Burnham Prebys Medical Discovery Institute, Orlando, United States; [2]Structural Biology Center, Biosciences Division, Argonne National Laboratory, Argonne, United States

*For correspondence: frastinejad@SBPdiscovery.org

Present address: [†]State Key Laboratory of Microbial Technology, School of Life Sciences, Shandong University, Qingdao, China

Competing interests: The authors declare that no competing interests exist.

**Abstract** The neuronal PAS domain proteins NPAS1 and NPAS3 are members of the basic helix-loop-helix-PER-ARNT-SIM (bHLH-PAS) family, and their genetic deficiencies are linked to a variety of human psychiatric disorders including schizophrenia, autism spectrum disorders and bipolar disease. NPAS1 and NPAS3 must each heterodimerize with the aryl hydrocarbon receptor nuclear translocator (ARNT), to form functional transcription complexes capable of DNA binding and gene regulation. Here we examined the crystal structures of multi-domain NPAS1-ARNT and NPAS3-ARNT-DNA complexes, discovering each to contain four putative ligand-binding pockets. Through expanded architectural comparisons between these complexes and HIF-1α-ARNT, HIF-2α-ARNT and CLOCK-BMAL1, we show the wider mammalian bHLH-PAS family is capable of multi-ligand-binding and presents as an ideal class of transcription factors for direct targeting by small-molecule drugs.

## Introduction

The mammalian bHLH-PAS transcription factors share a common protein architecture consisting of a conserved bHLH DNA-binding domain, tandem PAS domains (PAS-A and PAS-B), and a variable activation or repression domain. These factors can be grouped into two classes based on their heterodimerization patterns (*McIntosh et al., 2010*; *Bersten et al., 2013*) (*Figure 1A* and *Figure 1—figure supplement 1*). Class I includes the three hypoxia-inducible factor (HIF)-α proteins (HIF-1α, HIF-2α and HIF-3α), four neuronal PAS domain proteins (NPAS1-4), aryl hydrocarbon receptor (AHR), AHR repressor (AHRR), single-minded proteins (SIM1, SIM2) and clock circadian regulator (CLOCK); while class II includes aryl hydrocarbon receptor nuclear translocator (ARNT, also called HIF-1β), ARNT2, brain and muscle ARNT-like protein 1 (BMAL1, also called ARNTL) and BMAL2 (ARNTL2). Heterodimerization between class I and class II members produces functional transcription factors capable of DNA binding and target gene regulation.

Many bHLH-PAS proteins are significantly involved in human disease processes and would be ideal drug targets if their architectures could accommodate binding and modulation by small-molecules. Members of the HIF-α subgroup mediate the response to hypoxia and regulate key genetic programs required for tumor initiation, progression, invasion and metastasis (*Semenza, 2012*). AHR controls T cell differentiation (*Stevens and Bradfield, 2008*) and maintains intestinal immune homeostasis (*Leavy, 2011*), making it a potential target for alleviating inflammation and autoimmune diseases. Loss-of-function mutations in *SIM1* have been linked to severe obesity in human populations (*Bonnefond et al., 2013*), and defects in *SIM2* are associated with cancers (*Bersten et al.,*

**eLife digest** Transcription factors are proteins that can bind to DNA to regulate the activity of genes. One family of transcription factors in mammals is known as the bHLH-PAS family, which consists of sixteen members including NPAS1 and NPAS3. These two proteins are both found in nerve cells, and genetic mutations that affect NPAS1 or NPAS3 have been linked to psychiatric conditions in humans. Therefore, researchers would like to discover new drugs that can bind to these proteins and control their activities in nerve cells. Understanding the three-dimensional structure of a protein can aid the discovery of small molecules that can bind to these proteins and act as drugs. Proteins in the bHLH-PAS family have to form pairs in order to bind to DNA: NPAS1 and NPAS3 both interact with another bHLH-PAS protein called ARNT, but it is not clear exactly how this works.

In 2015, a team of researchers described the shapes that ARNT adopts when it forms pairs with two other bHLH-PAS proteins that are important for sensing when oxygen levels drop in cells. Here, Wu et al. – including many of the researchers involved in the earlier work – have used a technique called X-ray crystallography to determine the three-dimensional shapes of NPAS1 when it is bound to ARNT, and NPAS3 when it is bound to both ARNT and DNA. The experiments show that each of these structures contains four distinct pockets that certain small molecules might be able to bind to. The NPAS1 and NPAS3 structures are similar to each other and to the previously discovered bHLH-PAS structures involved in oxygen sensing.

Further analysis of other bHLH-PAS proteins suggests that all the members of this protein family are likely to be able to bind to small molecules and should therefore be considered as potential drug targets. The next step following on from this work is to identify small molecules that bind to bHLH-PAS proteins, which will help to reveal the genes that are regulated by this family. In the future, these small molecules may have the potential to be developed into new drugs to treat psychiatric conditions and other diseases in humans.

*2013*). CLOCK and BMAL1 together establish molecular circadian rhythms and their functional disruption can lead to a variety of metabolic diseases (*Gamble et al., 2014*).

The *NPAS* genes are highly expressed in the nervous system (*Zhou et al., 1997*; *Brunskill et al., 1999*; *Ooe et al., 2004*). In mice, genetic deficiencies in *NPAS1* and *NPAS3* are associated with behavioral abnormalities including diminished startle response, social recognition deficit and locomotor hyperactivity (*Erbel-Sieler et al., 2004*; *Brunskill et al., 2005*). NPAS2 is highly related to CLOCK in protein sequence (*Reick et al., 2001*), and altered patterns of sleep and behavioral adaptability have been observed in NPAS2-deficient mice (*Dudley et al., 2003*). *NPAS4* deficiency is associated with impairment of long-term contextual memory formation (*Ramamoorthi et al., 2011*). In humans, alterations in all four *NPAS* genes have been linked to neuropsychiatric diseases including schizophrenia, autism spectrum disorders, bipolar disease and seasonal depression disorders (*Kamnasaran et al., 2003*; *Pieper et al., 2005*; *Partonen et al., 2007*; *Pickard et al., 2009*; *Huang et al., 2010a*; *Bersten et al., 2014*; *Stanco et al., 2014*). Structural information has not been available for any NPAS proteins to show if they could bind drug-like molecules for treating psychiatric diseases.

A crystal structure was previously reported for the CLOCK-BMAL1 heterodimer (*Huang et al., 2012*), and we recently reported crystal structures for both HIF-2α-ARNT and HIF-1α-ARNT heterodimers (*Wu et al., 2015*). In all these complexes, the conserved bHLH-PAS-A-PAS-B protein segments were visualized. While no internal cavities were reported within the CLOCK-BMAL1 architecture; we identified multiple hydrophobic pockets within HIF-1α-ARNT and HIF-2α-ARNT heterodimers. Discrete pockets were encapsulated within each of the four PAS domains of their heterodimers (two within ARNT and two within each HIF-α protein) (*Wu et al., 2015*). Beyond the first structural characterizations of NPAS1-ARNT and NPAS3-ARNT complexes presented here, we further addressed if ligand-binding cavities are a common feature of mammalian bHLH-PAS proteins. A comparison of these two structures with those of CLOCK-BMAL1, HIF-1α-ARNT and HIF-2α-ARNT heterodimers unveils the larger mammalian bHLH-PAS family as ligand binding transcription factors

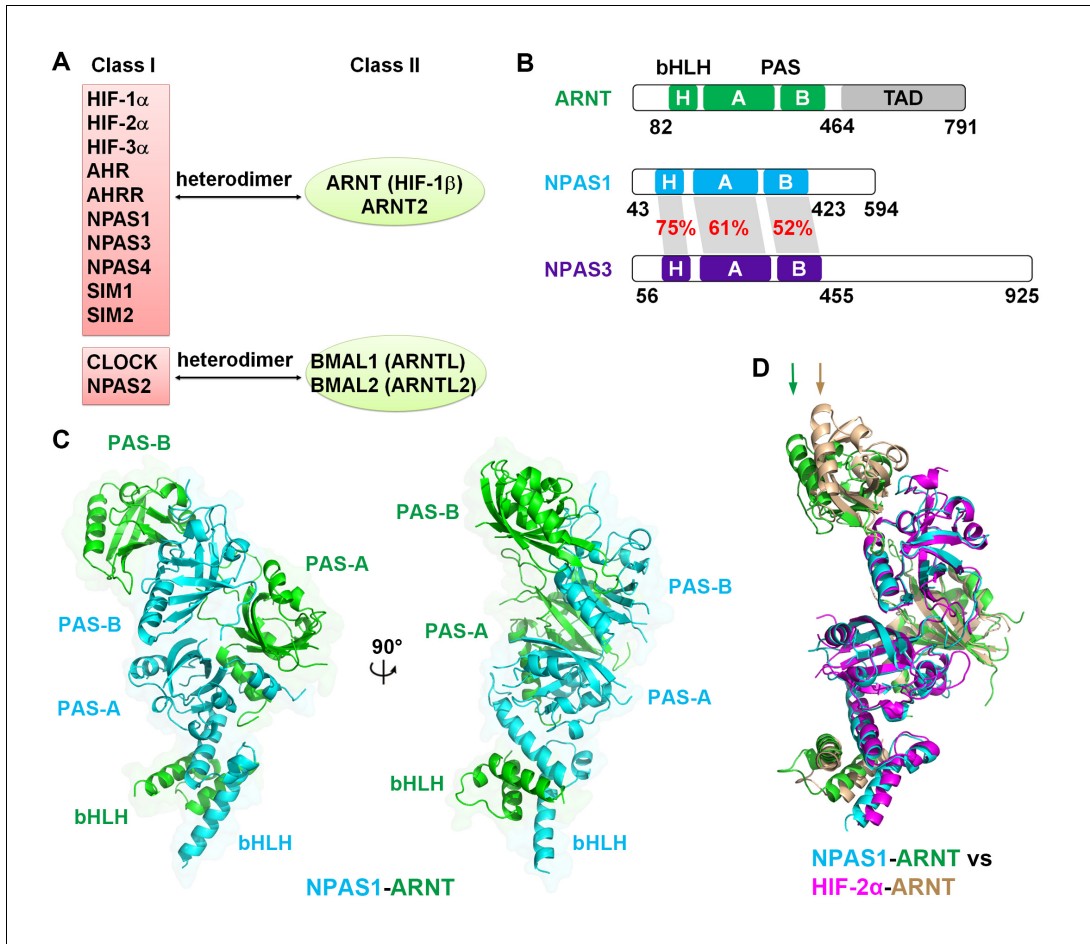

**Figure 1.** Comparison of bHLH-PAS proteins. (**A**) Heterodimerization patterns between bHLH-PAS proteins. (**B**) Protein domain arrangements of ARNT, NPAS1 and NPAS3. Percent amino-acid identities between corresponding domains of NPAS1 and NPAS3 are in red. (**C**) Overall crystal structure of the NPAS1-ARNT complex shown in two views. (**D**) Superposition of NPAS1-ARNT and HIF-2α-ARNT heterodimers. The arrows on top show the shift in position for the PAS-B domain of ARNT.

The following figure supplements are available for figure 1:

**Figure supplement 1.** Phylogenetic tree of all mouse bHLH-PAS family members based on protein sequences at their bHLH-PAS-A-PAS-B regions.

**Figure supplement 2.** Comparative protein sequence analysis of mouse bHLH-PAS proteins.

**Figure supplement 3.** Comparison of the overall structures of NPAS1-ARNT and CLOCK-BMAL1 complexes.

with internal pockets appropriate for the selective binding of lipophilic molecules and drug-like compounds.

## Results

### NPAS1-ARNT and NPAS3-ARNT architectures

We employed the contiguous bHLH-PAS-A-PAS-B segments of NPAS1, NPAS3 and ARNT proteins for our crystallographic studies (*Figure 1B*). For the NPAS1-ARNT heterodimer, we obtained crystals that diffracted to 3.2 Å resolution (*Table 1*). The quaternary organization of the NPAS1-ARNT

**Table 1.** Data collection and refinement statistics.

| | NPAS-ARNT | NPAS3-ARNT-DNA |
|---|---|---|
| **Data collection** | | |
| Space group | P 1 | P 43 |
| Cell dimensions | | |
| $a, b, c$ (Å) | 69.9, 81.2, 138.1 | 64.8, 64.8, 249.1 |
| $\alpha, \beta, \gamma$ (°) | 90.4, 95.1, 107.4 | 90.0, 90.0, 90.0 |
| Resolution (Å) | 50.0–3.20 (3.26–3.20)[*] | 50.0–4.20 (4.27–4.20) |
| $R_{merge}$ | 5.5 (76.4) | 6.0 (84.8) |
| CC* (highest resolution shell) | 0.793 | 0.971 |
| CC1/2 (highest resolution shell) | 0.459 | 0.893 |
| $I/\sigma I$ | 14.0 (1.2) | 20.0 (1.1) |
| Completeness (%) | 98.6 (98.3) | 94.6 (72.3) |
| Redundancy | 2.1 (2.2) | 5.2 (3.9) |
| | | |
| **Refinement** | | |
| Resolution (Å) | 37.7–3.20 (3.32–3.20) | 36.9–4.20 (5.28–4.20) |
| No. reflections | 39,096 (802) | 5868 (2120) |
| $R_{work}$/ $R_{free}$ (%) | 19.2/24.9 (28.3/40.4) | 29.5/36.2 (27.1/34.8) |
| No. atoms | | |
| Protein/DNA | 13,303 | 5313 |
| Water | 0 | 0 |
| B-factors | | |
| Protein/DNA | 49.6 | 66.9 |
| Water | - | - |
| R.m.s deviations | | |
| Bond lengths (Å) | 0.016 | 0.004 |
| Bond angles (°) | 1.46 | 0.77 |

One crystal was used for each structure.

[*]Highest resolution shell is shown in parenthesis.

complex is shown in *Figure 1C*. We found that the bHLH, PAS-A and PAS-B domains of ARNT twist along the outside surface of the NPAS1 protein. *Figure 1D* shows that NPAS1-ARNT and HIF-2α-ARNT heterodimers are very similar in overall architectures, but the PAS-B domain of ARNT is slightly displaced in the NPAS1 heterodimeric complex. This observation indicates that the ARNT architecture can display flexibility in accommodating its different class I partners.

We could not obtain crystals of *apo* NPAS3-ARNT and instead pursued its DNA complex. The response element for NPAS1-ARNT is known to match the consensus hypoxia response element (HRE) (*Ohsawa et al., 2005*; *Teh et al., 2007*), but the response element for NPAS3-ARNT was not previously characterized. NPAS1 and NPAS3 closely share amino-acids within their bHLH domain (75% identity, see *Figure 1B* and *Figure 1—figure supplement 2*), including conservation of residues that recognize DNA base-pairs based on our observations of the HIF-2α-ARNT-DNA complex (*Wu et al., 2015*). Therefore, we tested if NPAS3-ARNT could efficiently bind to the same consensus HRE sequence used by NPAS1-ARNT and HIF-α-ARNT heterodimers. We measured the dissociation constants ($K_d$) using a DNA duplex containing a central HRE sequence (5'-TACGTG-3') and found similar $K_d$ values of ~20 nM for both NPAS1-ARNT and NPAS3-ARNT (*Figure 2A*). These binding constants indicate a relatively higher affinity than that of the HIF-2α-ARNT heterodimer ($K_d$~40 nM) (*Wu et al., 2015*).

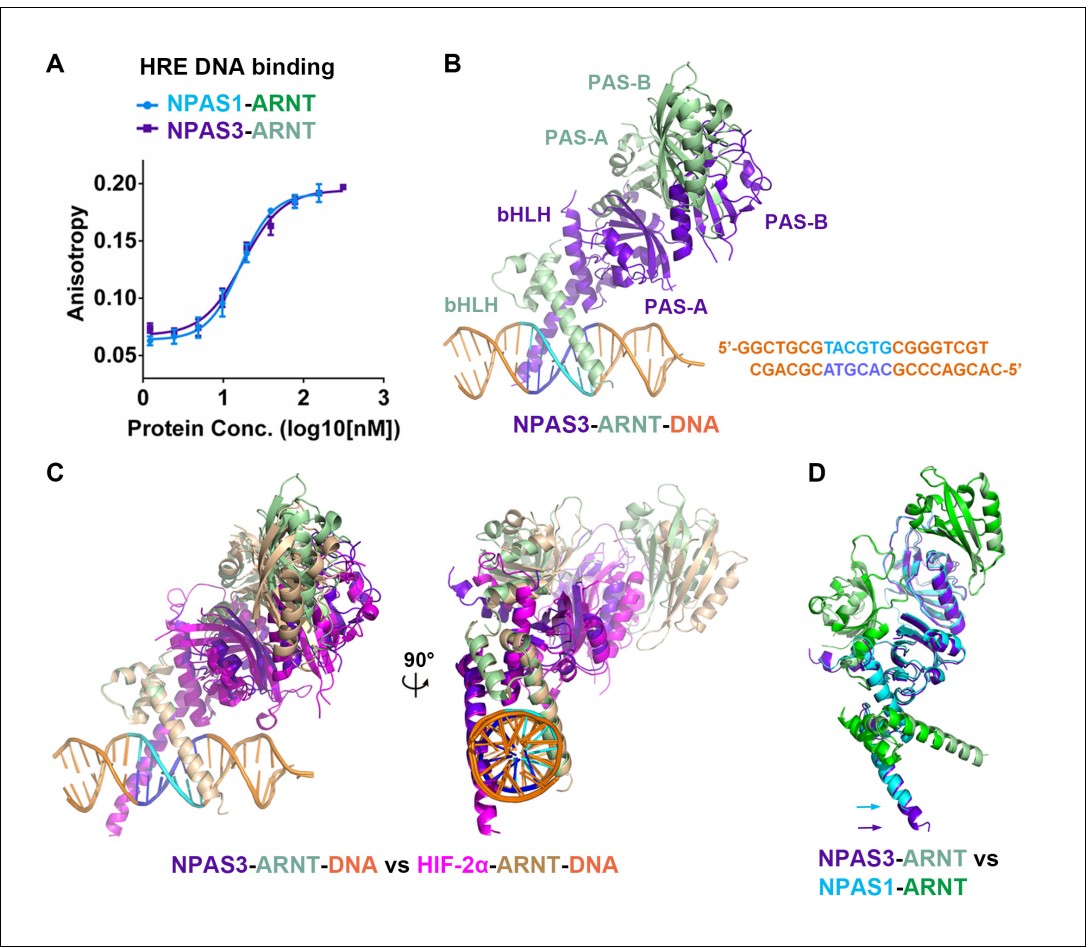

**Figure 2.** Comparison of NPAS1-ARNT and NPAS3-ARNT complexes. (**A**) DNA-binding affinities measured using fluorescence anisotropy. The $K_d$ values of NPAS1-ARNT and NPAS3-ARNT binding to the same HRE element were $16.3 \pm 1.1$ nM and $17.5 \pm 1.1$ nM, as calculated from three technical replicates respectively. (**B**) Overall structures of the NPAS3-ARNT-DNA, with the hexameric HRE site colored in cyan and blue. (**C**) Structure comparison of NPAS3-ARNT-DNA and HIF-2α-ARNT-DNA complexes aligned on their DNA. (**D**) Superposition of NPAS3-ARNT and NPAS1-ARNT structures. DNA was omitted from the NPAS3-ARNT complex. Arrows show the extension of α1 helices likely associated with its DNA binding.

We then obtained crystals for NPAS3-ARNT bound to this HRE element and solved the structure at 4.2 Å resolution (*Figure 2B* and *Table 1*). The resolution made possible an architectural comparison, at the quaternary level, between the NPAS3-ARNT, NPAS1-ARNT, HIF-1α-ARNT and HIF-2α-ARNT heterodimers. All of these complexes share a similar overall architecture (*Figure 2C,D*) stabilized by the same six domain-domain junctions (see below). Furthermore, the cooperation between the two subunits of NPAS3-ARNT also creates a DNA-reading head that is similar to that of the HIF-α-ARNT-DNA complexes, allowing direct readout of the HRE site (*Figure 2C*).

## Domain-domain arrangements

We next tested if the six domain-domain interfaces observed in NPAS1-ARNT and NPAS3-ARNT (*Figure 3A*) are important for maintaining the stability of their full-length heterodimers within cells. For our study, we used co-immunoprecipitation (co-IP) experiments in HEK293T cells, together with a series of single or double mutations positioned within ARNT (*Figure 3B*). Mutations in interfaces 1–4 were found to significantly destabilize ARNT's heterodimeric interactions with both NPAS1 and NPAS3, as predicted from their crystal structures. Point mutations within NPAS1 at interfaces 5 and 6 also destabilized heterodimerization with ARNT (*Figure 3C*). We additionally tested the effects of

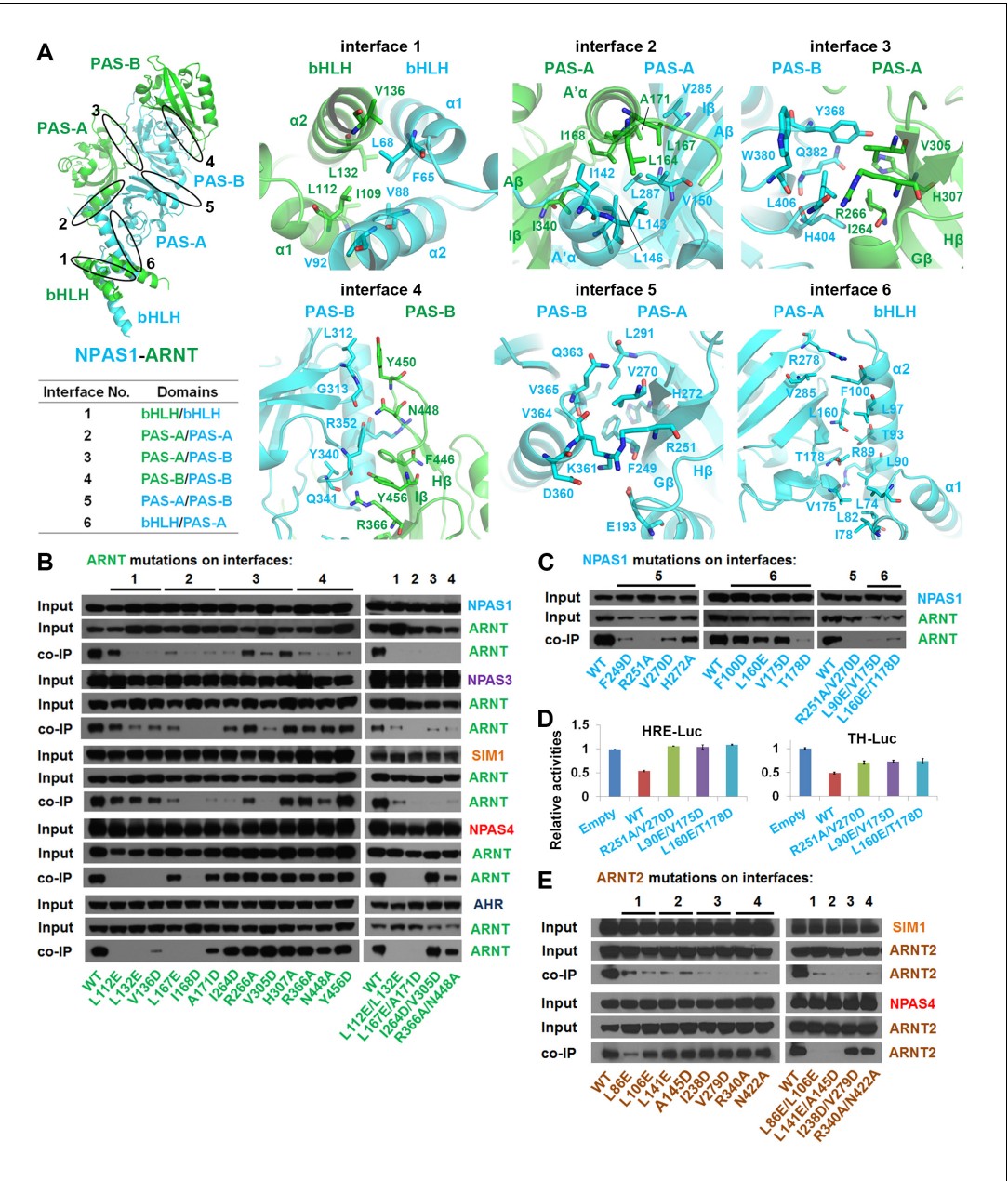

**Figure 3.** Testing of domain interfaces in NPAS1-ARNT and other ARNT heterodimers using mutagenesis. (**A**) Detailed interactions at each of six interfaces in NPAS1-ARNT. Each junction is circled in the context of the overall structure of NPAS1-ARNT on the left, and each interface is further listed in the table. (**B**) Co-IP experiments showing the effects of ARNT mutations at interfaces 1–4 on the cellular stabilities of heterodimers formed with NPAS1, NPAS3, SIM1, NPAS4 and AHR, respectively. (**C**) Co-IP experiments showing the effects of NPAS1 mutations at interfaces 5 and 6 on the stabilities of NPAS1-ARNT heterodimer. (**D**) Luciferase reporter assay testing the effects of NPAS1 (wide-type and three mutants) on HRE-driven (left) and tyrosine hydroxylase (TH) promoter-driven (right) transactivation. Each sample represents the average reading of cells from three wells. (**E**) Co-IP experiments showing the effects of ARNT2 mutations at interfaces 1–4 (corresponding to ARNT) on the stabilities of heterodimers formed with SIM1 and NPAS4. The residues and mutants of ARNT, NPAS1 and ARNT2 are labelled in green, cyan and brown, respectively.

The following figure supplement is available for figure 3:

**Figure supplement 1.** Amino-acid sequence alignment of full-length mouse ARNT and ARNT2 proteins.

these destabilizing mutations on the transcriptional function of NPAS1-ARNT. NPAS1 has been shown to function as a transcriptional repressor on its target gene *tyrosine hydroxylase (TH)*, since it lacks a functional transactivation domain (*Teh et al., 2006*). We confirmed this repression activity in both HRE-driven and TH-driven reporter assays, and further found that mutations that destabilized the heterodimer also compromised the transcriptional repression (*Figure 3D*).

In discovering that the NPAS1-ARNT, NPAS3-ARNT and HIF-α-ARNT heterodimers utilize the same six domain-domain interfaces to create a shared quaternary structure (*Figures 1D* and *2D*), we also found that this type of architecture is clearly distinct from that of CLOCK-BMAL1 complex (*Huang et al., 2012*) (*Figure 1—figure supplement 3*). Therefore, we conclude that there are at least two distinct architectural forms within the mammalian bHLH-PAS family. Since the ARNT heterodimers form a larger group than the BMAL1 heterodimers (*Figure 1A*), we asked if other ARNT heterodimers rely on the same six domain-domain junctions we observed in NPAS1-ARNT, NPAS3-ARNT and HIF-α-ARNT heterodimers. We could experimentally interrogate the degree of architectural variation within the ARNT heterodimer class by using our panel of ARNT mutations that were able to block its heterodimerization with NPAS1 and NPAS3.

Using co-IP studies, we tested if these same ARNT mutations would disrupt its heterodimeric complexes with SIM1, NPAS4 and AHR (*Figure 3B*). The SIM1-ARNT heterodimer stability was indeed compromised by these ARNT mutations, indicating that this heterodimer shares the same overall architecture as NPAS1-ARNT, NPAS3-ARNT and the two HIF-α-ARNT heterodimers. However, NPAS4-ARNT and AHR-ARNT heterodimer stabilities were not impacted in the same manner by these ARNT mutations, particularly when the mutations were located to interfaces 3 and 4. Therefore, we believe there is greater architectural variation within ARNT heterodimer group than what has been crystallographically observed to date. A protein amino-acid sequence alignment further indicates that residues observed to stabilize domain-domain junctions in the NPAS1/3-ARNT and HIF-α-ARNT heterodimers are much more conserved in SIM1 than in NPAS4 and AHR (*Figure 1— figure supplements 1* and *2*).

The ARNT protein is ubiquitously expressed in mammalian cells, but its paralog ARNT2 is more specifically enriched in brain and kidney tissues (*Hirose et al., 1996*). ARNT and ARNT2 have both unique and overlapping cellular functions (*Keith et al., 2001*), and ARNT2 has been suggested as the preferred physiological partner for Class I members SIM1 (*Michaud et al., 2000*) and NPAS4 (*Bersten et al., 2014*). Since the amino-acid sequence identity at the bHLH-PAS region is nearly 80% between ARNT and ARNT2, and since all the ARNT residues observed to be participating at dimerization interfaces are fully conserved in ARNT2 (*Figure 3—figure supplement 1*), we predict that ARNT2 heterodimers will display similar overall quaternary architectures as their ARNT counterparts. To test this prediction, we mutated two ARNT2 residues at each of the four dimer interfaces (*Figure 3E*). Compared with the ARNT mutations (*Figure 3B*), these corresponding ARNT2 mutations (both single and double ones) indeed had similar effects on the dimerization with not only SIM1 but also NPAS4 (*Figure 3E*), indicating that ARNT2 indeed dimerizes with SIM1 and NPAS4 in the same way as ARNT does.

## NPAS1-ARNT and NPAS3-ARNT cavities

In both HIF-1α-ARNT and HIF-2α-ARNT heterodimers, we previously identified hydrophobic pockets encapsulated within the two PAS domains of ARNT, and within the two PAS domains of each HIF-α protein (*Wu et al., 2015*). Here we asked if NPAS1-ARNT and NPAS3-ARNT harbored similarly positioned pockets. NPAS1 and NPAS3 are genetically associated with a wide range of human neuropsychiatric disorders (*Kamnasaran et al., 2003*; *Pieper et al., 2005*; *Stanco et al., 2014*). Thus, the discovery of ligand-binding cavities could lead to the future discovery of therapeutic molecules for these illnesses. We show that NPAS1 protein's PAS-A and PAS-B domains do contain internal cavities with volumes measuring 190 $\text{Å}^3$ and 180 $\text{Å}^3$, respectively (*Figure 4A,B* and *Table 2*). Similarly positioned pockets are seen in the NPAS3 protein, measuring 100 $\text{Å}^3$ and 230 $\text{Å}^3$, respectively. Each of these PAS domains resembles a baseball catcher's mitt, with the beta strands forming the palm and short alpha-helices forming the opposing thumb to enclose a central pocket.

Through heterodimerization, ARNT further brings its own two pockets to join each of its class I partners. We examined if the two ARNT pockets alter their shape when ARNT forms different heterodimeric complexes. We could not detect any major change in the cavity size or shape of ARNT's PAS-A and PAS-B pockets, whether this protein was in a complex with HIF-2α or NPAS1

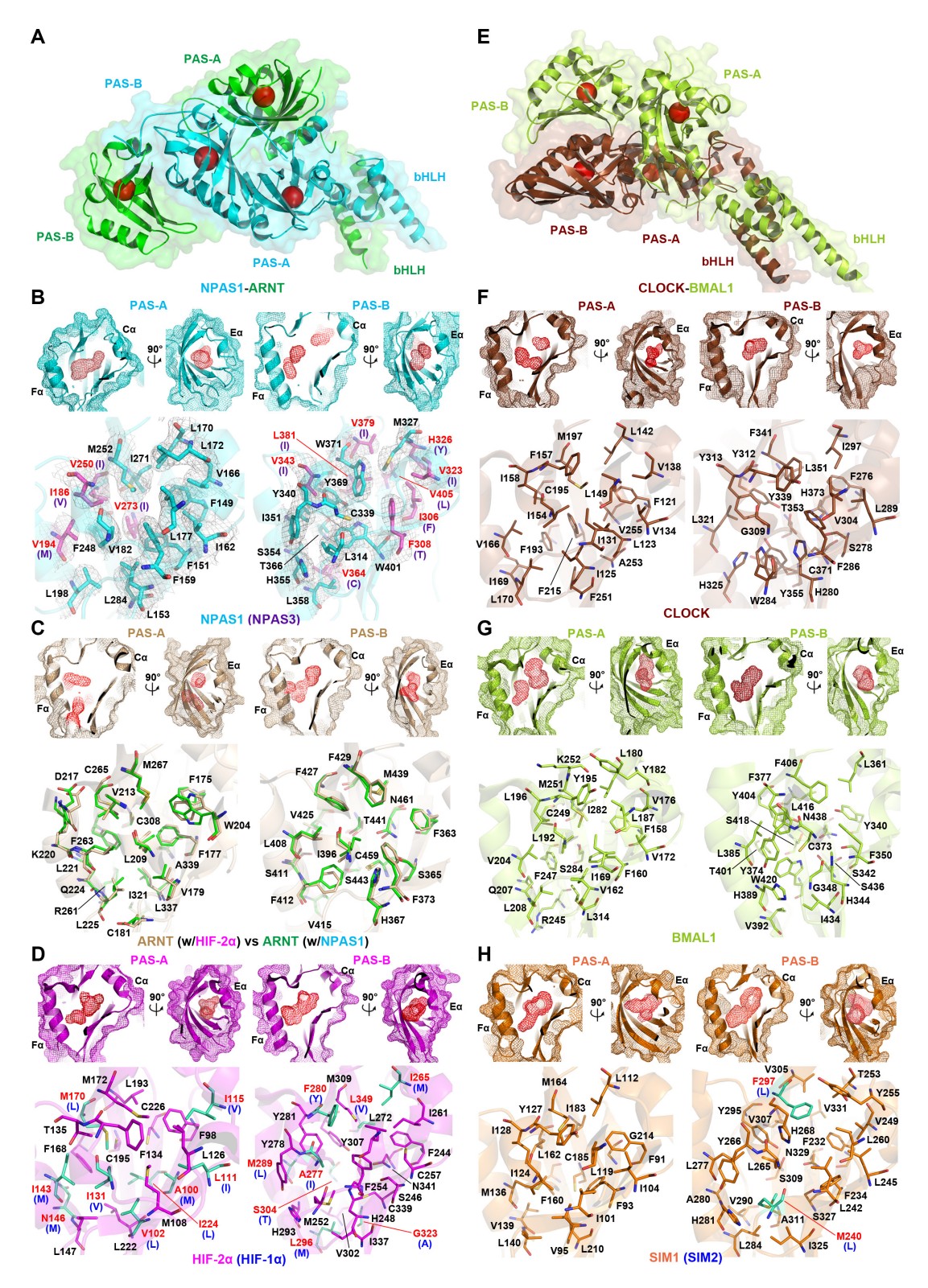

**Figure 4.** Comparison of ligand-binding pockets among multiple bHLH-PAS proteins. (A and E) Relative positions of the four pockets (red circles) within the structures of NPAS1-ARNT (A) and CLOCK-BMAL1 (E) complexes. (B–D and F–H) Empty pockets in each of PAS-A and PAS-B domains of NPAS1 (B), ARNT (C), HIF-2α (D), CLOCK (F), BMAL1 (G) and SIM1 (H). The accessible cavities are shown in red meshes, together with the amino-acid residues lining each pocket. For NPAS1, the surrounding residues are further covered with $2F_{O-}F_{C}$ map contoured at 1.0σ (B). The NPAS1 residues

*Figure 4 continued on next page*

*Figure 4 continued*

with identical counterparts in NPAS3 are in cyan and labelled in black, while those with non-conserved NAPS3 counterparts (indicated in parentheses) are in magenta and labelled in red (**B**). ARNT pocket residues from *apo* HIF-2α-ARNT complex (PDB: 4ZP4) (in wheat) and those from NPAS1-ARNT complex (in green) are superposed for comparison (**C**). The HIF-2α residues with identical counterparts in HIF-1α are in magenta, while those with non-conserved HIF-1α counterparts (indicated in parentheses) are in green (**D**). Structures of CLOCK (**F**) and BMAL1 (**G**) are from the CLOCK-BMAL1 complex (PDB: 4F3L) in brown and lime, respectively. The SIM1 structure in orange was modelled from NPAS1 and HIF-2α; and the pocket residues not conserved in SIM2 are in green with their SIM2 counterparts indicated in parentheses (**H**).

(*Figure 4C*). ARNT is ubiquitously expressed in many cell types and is predominantly nuclear (*Reyes et al., 1992*); whereas its class I heterodimerization partners are signal activated and/or tissue restricted. Therefore, the pockets located in NPAS1, NPAS3, HIF-1α and HIF-2α should allow more selective actions of therapeutic drugs than the pockets within ARNT (*Figure 4B–D*).

## Variations in protein pockets

Our findings of two similarly positioned pockets within each of NPAS1, NPAS3, HIF-1α, HIF-2α and ARNT led us to consider if any other bHLH-PAS family members could also harbor putative ligand-binding cavities. The crystal structure of CLOCK-BMAL1 has been reported, but no internal cavities were described in its published analysis (*Huang et al., 2012*). Therefore, we closely examined this structure and found that both CLOCK and BMAL1 proteins contained discrete pockets within each of their PAS-A and PAS-B domains (*Figure 4E–G*). The BMAL1's PAS-A and PAS-B pockets (measuring 200 Å$^3$ and 220 Å$^3$, respectively) are larger than those of CLOCK (measuring 120 Å$^3$ and 140 Å$^3$, respectively) (*Table 2*). Crystal structures are unavailable for SIM1 and SIM2 proteins; however, the close evolutionary and amino-acid sequence similarity with the NPAS1 protein led us to generate plausible models for their PAS-A and PAS-B pockets (*Figure 4H*, *Figure 1—figure supplements 1* and *2*).

The pockets of AHR and its repressor AHRR are more difficult to model based on our existing crystal structures, because of greater sequence and evolutionary divergence of these proteins (*Figure 1—figure supplement 1*). Interestingly, multiple classes of ligands, including halogenated aromatic hydrocarbons, tetrapyrroles and several tryptophan derivatives were previously identified as direct AHR binding ligands (*Denison et al., 2011*; *Stejskalova et al., 2011*). These molecules display significant variations in their chemical structures and sizes, but still bind with high affinities to AHR ($K_d$ values of 0.1–100 nM). High-affinity, multi-ligand binding can best be accounted for by the use of four discrete pockets within AHR-ARNT, as shown here for other ARNT heterodimers, than just one pocket as suggested previously (*Pandini et al., 2007*).

*Figure 4* shows side-by-side comparisons of the PAS-A and PAS-B pockets from multiple bHLH-PAS proteins. Importantly, each PAS domain relies on a constellation of different amino-acids to form its interior cavity, allowing the pockets to bind selectively to different repertoires of endogenous ligands. Moreover, within subgroups such as HIF-1α/2α, NPAS1/3, and SIM1/2, the proteins share pocket residues more closely with each other than they do between subgroups. This observation indicates close ligand preferences within subgroups, suggesting they could recognize similar

**Table 2.** Volumes of ligand-binding pockets of bHLH-PAS proteins.

| Location | NPAS1 | NPAS3 | HIF-1α | HIF-2α | HIF-3α | ARNT | CLOCK | BMAL1 | NPAS2 | SIM1 | SIM2 |
|---|---|---|---|---|---|---|---|---|---|---|---|
| PAS-A | 190 | 100 | 100 | 150 | 170 | 110 | 120 | 200 | 230 | 210 | 210 |
| PAS-B | 180 | 230 | 160 | 370 | 590 | 210 | 140 | 220 | 170 | 370 | 310 |

Pocket volumes (Å$^3$) were calculated using CASTp program (*Dundas et al., 2006*) using the default probe sphere radius of 1.4 Å. The PDB coordinate files used for NPAS1 and NPAS3 proteins were from the NPAS1-ARNT and NPAS3-ARNT-DNA complexes, the coordinates for ARNT and HIF-2α were from the *apo* HIF-2α-ARNT complex (PDB: 4ZP4), and the coordinates for CLOCK and BMAL1 were from the CLOCK-BMAL1 complex (PDB: 4F3L). The values of PAS-B domains of HIF-1α and HIF-3α (fatty acid bound) were from high-resolution single domain structures (PDBs: 4H6J and 4WN5), respectively. The PAS-A domain of HIF-3α, and both PAS domains of NPAS2, SIM1 and SIM2 were modeled using the SWISS-MODEL server (*Biasini et al., 2014*).

metabolites or signaling molecules derived from the same biosynthetic pathway. Importantly, in all the PAS-A and PAS-B pockets, the amino-acid residues lining interior cavities are predominantly hydrophobic. This property would allow favorable van der Waals interactions with lipophilic ligands. The desolvation of hydrophobic ligands can further contribute to favorable energetics of ligand-binding inside these cavities.

The internal pocket volumes in the bHLH-PAS family (100–600 $\text{Å}^3$) (*Table 2*) are in-line with pocket sizes observed in other classes of ligand-binding proteins (100–1000 $\text{Å}^3$) (*Liang et al., 1998*). Moreover, we found that ligand binding can increase the size of pocket significantly. For example, HIF-2α specific inhibitor 0X3 enlarged its PAS-B pocket volume from 370 $\text{Å}^3$ to 560 $\text{Å}^3$ (*Wu et al., 2015*). These observations suggest a high degree of adaptability in PAS domain pockets, and should encourage future drug-discovery efforts seeking multiple classes of synthetic modulators for bHLH-PAS proteins. An analogy can be made with the nuclear receptor family, where diverse classes of small-molecules can bind to the same pocket, and ligand binding can expand pocket sizes (*Huang et al., 2010b*).

## Discussion

Previously, the nuclear receptors were believed to form the only transcription factor family in higher eukaryotes with conserved ligand binding capabilities. We showed here that the mammalian bHLH-PAS proteins constitute a second independent family of transcription factors with the appropriate and conserved molecular frameworks required for multi-ligand binding. Our findings are based on the direct crystallographic analysis of seven distinct bHLH family members (ARNT, HIF-1α, HIF-2α, NPAS1, NPAS3, CLOCK and BMAL1). In all these proteins, internal hydrophobic cavities were clearly observed within both their PAS-A and PAS-B domains. These PAS domains become interweaved in bHLH-PAS heterodimers through at least two highly distinct forms of architecture. Aside from the seven members we crystallographically examined, three additional members: AHR (*Stejskalova et al., 2011*), HIF-3α (*Fala et al., 2015*) and NPAS2 (*Dioum et al., 2002*) are known to have ligand-binding capabilities associated with at least one of their PAS domains, bringing the experimentally confirmed number of ligand pocket containing members to ten. Our sequence-structure comparative analyses further implicate three more members: ARNT2, SIM1 and SIM2, as likely to harbor pockets due to considerable amino-acid conservations with ARNT and NPAS1.

Our unmasking of the wider bHLH-PAS protein family as a group of transcription factors with multi-ligand-binding capabilities should fuel the future search for their physiological and endogenous ligands. The protein pockets, in each case, make these factors ideally suited for applications of high-throughput screening campaigns to find small-molecule therapeutics for a variety of human diseases, including psychiatric illnesses, cancers and metabolic diseases. The distinctive amino-acid residues inside their pockets predict successful outcomes for screens seeking highly selective modulators for each protein. These pockets are also highly pliable and can accommodate significantly larger ligands than their empty volumes alone suggest.

The precise mechanism through which endogenous and/or pharmacologic ligands can modulate transcriptional activities of bHLH-PAS proteins has not yet been fully revealed, and could differ for each family member. For example, ligand binding to AHR can displace heat shock protein 90 and initiate the nuclear translocation of AHR (*Soshilov and Denison, 2011*). Some small-molecule HIF-2α antagonists that bind to the PAS-B domain can disrupt the dimerization between HIF-2α and ARNT (*Scheuermann et al., 2013*). Beyond these initial observations to date, and without the availability of small-molecule tools to probe other bHLH-PAS proteins, the mechanistic links between ligand binding and transcriptional regulation remain to be discovered.

It is interesting that while dimeric nuclear receptors harbor two pockets in their functional architectures, the bHLH-PAS proteins present four distinct pockets. Notably a fifth pocket was also observed in HIF-2α-ARNT heterodimers, located between two subunits, allowing proflavine to promote subunit dissociation (*Wu et al., 2015*). The latter finding further suggests that crevices formed between the PAS domains within the quaternary architectures of family members could form additional ligand binding and modulation sites. The availability of so many distinctive pockets within bHLH-PAS proteins could allow a variety of biosynthetic or metabolic signals derived from cellular pathways to be integrated into a single unified functional response. Alternatively, each pocket may allow a bHLH-PAS protein to be the site of unique ligand within each cell-type. The identification

and validation of endogenous ligands for this family will help define the genetic programs controlled by each bHLH-PAS member.

## Materials and methods

### Plasmid construction and site-directed mutagenesis

For protein overexpression in *Escherichia coli*, mouse NPAS1 (GenBank accession: AAI32114.1, residues 43–423) and NPAS3 (GenBank accession: AAI67248.1, residues 56–455) were cloned into the pSJ2 vector, respectively. For the co-immunoprecipitation studies, full-length mouse NPAS1, NPAS3, SIM1 and NPAS4 were cloned into the pCMV-Tag4 vector (C-terminal Myc-tagged), and mouse ARNT2 was cloned into the pCMV-Tag1 vector (C-terminal Flag-tagged). The cloning of ARNT and AHR constructs has been described previously (*Wu et al., 2013*, *2015*). Site-directed mutagenesis was confirmed in each case by DNA sequencing.

### Protein expression and purification

The recombinant plasmids pSJ2-NPAS1 and NPAS3 were co-transformed along with pMKH-ARNT into BL21-CodonPlus (DE3)-RIL competent cells (Agilent Technologies, Santa Clara, CA, #230245). Proteins were expressed and purified as previously described (*Wu et al., 2015*). To prepare NPAS3-ARNT DNA-bound complexes, synthetic 21mer double-strand DNA (forward: 5'- GGCTGCGTACG TGCGGGTCGT-3' and reverse: 5'-CACGACCCGCACGTACGCAGC-3') was mixed with the hetero-dimeric proteins.

### Crystallization and X-ray data collection

Crystallization of the NPAS1-ARNT complex was carried out using the sitting drop vapor diffusion method at 16°C, by mixing equal volume of protein (4 mg/ml) and reservoir solution containing 2% Tacsimate pH 7.0, 3% PEG3350. Before flash frozen in liquid nitrogen, crystals were soaked in reservoir plus 30% glycerol as the cryoprotectant. NPAS3-ARNT-DNA crystals were grown at 16°C in sitting drops formed by equal volume of complex (4 mg/ml) and reservoir consisting of 100 mM $NH_4F$, 9% PEG 3350, and then transferred stepwise to cryoprotectant of 100 mM $NH_4F$, 10% PEG 3350 and up to 30% PEG 400 (5% increase each step) prior to flash freezing. Diffraction data were collected at the Argonne National Laboratory SBC-CAT 19ID beamline at 100 K.

### Structure determination and refinement

The structures of NPAS1-ARNT and NPAS3-ARNT-DNA complexes were solved by molecular replacement with Phaser (RRID: SCR_014219) (*McCoy et al., 2007*), using the HIF-2α-ARNT structures (PDB: 4ZP4 and 4ZPK) as the search models. Further manual model building was facilitated using Coot (RRID: SCR_014222) (*Emsley et al., 2010*), combined with the structure refinement using Phenix (RRID: SCR_014224) (*Adams et al., 2010*). The diffraction data and final statistics are summarized in *Table 1*. The Ramachandran statistics, calculated by Molprobity (RRID: SCR_014226) (*Chen et al., 2010*), are 93/0.06% and 89/0.19% (favored/outliers) for NPAS1-ARNT and NPAS3-ARNT-DNA complexes, respectively. All the structural figures were prepared using PyMol (The Pymol Molecular Graphics System, RRID: SCR_000305). Coordinates and structure factors have been deposited in Protein Data Bank under accession numbers 5SY5 (NPAS1-ARNT) and 5SY7 (NPAS3-ARNT-DNA).

### Fluorescence polarization binding assay

The 21-mer fluoresceinated double-strand DNA was prepared by annealing 6-FAM labelled forward strand (5'-GGCTGCGTACGTGCGGGTCGT-3') with the unlabeled reverse strand (5'-ACGACCCG-CACGTACGCAGCC-3') in the buffer consisting of 10 mM Tris pH 7.5, 1 mM EDTA and 2 mM $MgCl_2$. For the binding assay, 2 nM DNA was incubated with purified proteins for 30 min, and final protein concentrations were varied by serial dilution in binding buffer (20 mM Tris pH 8.0, 50 mM NaCl and 10 mM DTT). The fluorescence polarization signals were recorded and processed as previously (*Wu et al., 2015*).

## Co-immunoprecipitation

HEK293T cells (ATCC CRL-3216, RRID: CVCL_0063) were seeded in 10 cm dishes and cultured in DMEM containing 10% FBS (Thermo Fisher Scientific, #11995 and #16000) at 37°C with 5% $CO_2$. One day later, cells were transfected with 2 μg pCMV-Tag4-NPAS1, NPAS3, SIM1, NPAS4 or AHR (WT or mutants) and 6 μg pCMV-Tag1-ARNT or ARNT2 (WT or mutants) plasmids using 16 μL jet-PRIME regent (Polyplus-transfection, #114–07). After overnight incubation, medium was refreshed (10 nM 2,3,7,8-tetrachlorodibenzo-*p*-dioxin added in the case of AHR). Another 24 hr later, cells were harvested and immunoprecipitation was performed similarly to our previous work (*Wu et al., 2015*).

## Luciferase reporter assay

HEK293T cells were seeded in 24-well plates, and one day later transfected with 200 ng of pCMV-Tag4-NPAS1 (WT, mutants or empty plasmid), 1 ng of pRL (control Renilla luciferase), 100 ng of HRE-luc reporter (*Ao et al., 2008*) or TH-luc reporter containing the tyrosine hydroxylase promoter sequence (*Thiel et al., 2005*) using 0.6 μL jetPRIME regent for each well. Medium was refreshed after overnight transfection, and luciferase activity was measured another 24 hr later using the Dual-Glo Luciferase Assay System (Promega, #E2920). Final data were normalized by the relative ratio of firefly and Renilla luciferase activity.

## Acknowledgement

We thank Chen-Ming Fan at Carnegie Institution for Science for the mouse SIM1 plasmid, Jianrong Lu at University of Florida for the HRE-luc reporter, Gerald Thiel at University of Saarland for the TH-luc reporter, members of the Structural Biology Center at Argonne National Laboratory for their help with data collection at the 19-ID beamline, Dominika Borek at UT Southwestern Medical Center and Jingping Lu at SBP for help with diffraction data processing during the 2014 CCP4/APS summer school. This work was supported by the National Institutes of Health (grant number NIGMS 1R01GM117013) and US ARMY Medical Research (grant number W81XWH-16–1-0322).

## Additional information

### Funding

| Funder | Grant reference number | Author |
| --- | --- | --- |
| National Institutes of Health | NIGMS 1R01GM117013 | Fraydoon Rastinejad |
| Army Research Office | W81XWH-16-1-0322 | Fraydoon Rastinejad |

The funders had no role in study design, data collection and interpretation, or the decision to submit the work for publication.

### Author contributions

DW, Conceived this study, Analyzed data, Wrote the manuscript, Purified the proteins, Carried out crystallizations, Solved the structures, Conducted biochemical and cell-based experiments; XS, Conducted biochemical and cell-based experiments; NP, Produced the expression and mutation constructs; YK, Collected and processed synchrotron diffraction data; FR, Conceived this study, Analyzed data and wrote the manuscript

### Author ORCIDs

Fraydoon Rastinejad, http://orcid.org/0000-0002-0784-9352

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
