## [Decision Letter]

Thank you for submitting your article "Structures of NPAS1-ARNT and NPAS3-ARNT Complexes Implicate bHLH-PAS Family as Multi-Ligand Binding Transcription Factor" for consideration by *eLife*. Your article has been favorably evaluated by Richard Aldrich as the Senior Editor and three reviewers, including Ronald M Evans (Reviewer #3) and a member of our Board of Reviewing Editors.

The reviewers have discussed the reviews with one another and the Reviewing Editor has drafted this decision to help you prepare a revised submission.

Summary:

This manuscript reports a structural and biochemical characterization of two heterodimeric bHLH-PAS domain proteins, NPAS1-ARNT and NPAS3-ARNT, that probes differences in domain-domain arrangements among different subclasses of PAS proteins and identifies broad conservation of ligand-binding pockets that appear capable of binding a broad array of ligands. The structure of NPAS3-ARNT bound to the hypoxia response element (HRE) recognized by NPAS1 establishes the specificity and provides mechanistic insight into DNA binding by NPAS3. Mutations in NPAS1 and NPAS3 underlie varies psychiatric disordered, making these potential targets of drug discovery. A comparison of these structures with a previous structure of related family members, CLOCK and BMAL1, reveals that the latter may also contain previously unnoticed ligand-binding pockets, thereby suggesting that bHLH-PAS proteins may be considered a new class of ligand-regulated transcription factors. The in vivo relevance of these structures is supported by transcriptional reporter assays and protein co-immunoprecipitation studies, where mutations at domain-domain interfaces identified in the structures disrupt the heterodimeric protein-protein interactions. Furthermore, the differential effects of the mutations in ARNT on its interactions with the additional bHLH-PAS family members SIM1, NPAS4 and AHR provide insight into its function as a common co-receptor.

Essential revisions:

1) Data shown in Figure 3 is based on HRE-luciferase reporters. This data should be replaced with validated NPAS/ARNT consensi from one or more of the promoters/enhancers of bona fide NPAS target genes.

2) Mutations found in ARNT that disrupt SIM1 and NPAS4 dimerization should be introduced into ARNT2, and the IPs repeated. ARNT2 is their physiological partner, and this should be confirmed in the 293T assays, not simply inferred from sequence alignment.

3) While the authors draw a comparison between bHLH-PAS proteins and nuclear receptors as ligand binding transcription factors, the conformational changes induced by ligand binding in PAS domains appear subtle, raising questions on how endogenous and/or pharmacologic ligands targeting these pockets affect transcriptional activity. Some discussion on potential mechanisms would improve the manuscript.

---

## [Author Response]

*Essential revisions:*

*1) Data shown in Figure 3 is based on HRE-luciferase reporters. This data should be replaced with validated NPAS/ARNT consensi from one or more of the promoters/enhancers of bona fide NPAS target genes.*

In addition to the HRE-Luc reporter assay, we have now included a new reporter assay in which the luciferase activity is under the control of tyrosine hydroxylase (known NPAS1 target gene) promoter sequence (shown in Figure 3).

*2) Mutations found in ARNT that disrupt SIM1 and NPAS4 dimerization should be introduced into ARNT2, and the IPs repeated. ARNT2 is their physiological partner, and this should be confirmed in the 293T assays, not simply inferred from sequence alignment.*

Following this suggestion, we have mutated corresponding ARNT2 residues at the interfaces and carried out a new set of co-IP experiments with SIM1 and NPAS4 (data shown in Figure 3 and discussed in the last paragraph of the subsection “Domain-domain arrangements”).

*3) While the authors draw a comparison between bHLH-PAS proteins and nuclear receptors as ligand binding transcription factors, the conformational changes induced by ligand binding in PAS domains appear subtle, raising questions on how endogenous and/or pharmacologic ligands targeting these pockets affect transcriptional activity. Some discussion on potential mechanisms would improve the manuscript.*

The reviewer has raised a very important and interesting question. A short paragraph is now included in the Discussion section about currently known mechanisms that relate ligand binding to changes in the functions of bHLH-PAS proteins (third paragraph). To more comprehensively explore the mechanisms that can relate ligand-binding to transcriptional regulation, one must first have in hand endogenous ligands (or synthetic tool compounds) for NPAS1, NPAS3 and other bHLH-PAS proteins. Unfortunately, almost all bHLH-PAS members remain “orphans” at this time, presenting a significant barrier to uncovering the full range of mechanisms that could relate ligand binding to changes in transcriptional activity.